# High-Speed Railway Opening and Corporate Fraud

Chen Wang [1], Jack Strauss [2] and Lei Zheng [1,*]

1　School of Economics and Management, Beijing Jiaotong University, Beijing 100044, China; 20113023@bjtu.edu.cn
2　Reiman School of Finance, University of Denver, 2101 S. University Blvd, Denver, CO 80208, USA; jack.strauss@du.edu
*　Correspondence: rayzhenglei@bjtu.edu.cn

**Abstract:** The impact of high-speed railway (HSR) on corporate behavior has recently attracted both practical and theoretical interest. In this paper, based on a sample of A-share listed companies from 2007 to 2020 in China, we use a difference-in-difference model to explore the impact of HSR openings on corporate fraud and analyze its mechanism. We find that HSR introduction has several important implications. First, it reduces the tendency and frequency of corporate fraud. Second, HSR opening restrains corporate fraud by improving the external supervision level and reducing the financing constraints of the company. Third, the inhibitory effect of the HSR opening on corporate fraud is significant when the market competition is less intense, and the company's internal control level is poor. Fourth, after distinguishing types of fraud, HSR opening can still significantly inhibit information disclosure fraud and manager fraud, but not operation fraud. These results indicate that HSR openings promote the flow of information and labor across regions, alleviating the information asymmetry of firms. Our findings are conducive to improving the governance environment of the listed companies, which provides new clues for discovering and restricting corporate fraud.

**Keywords:** high-speed railway opening; corporate fraud; geographical distance

## 1. Introduction

Since the 2008 opening of the Beijing Tianjin Intercity Railway, the HSR has developed rapidly in China. By 2020, the operating mileage of China's HSR has reached 37,900 km, covering nearly all cities with a population of more than 1 million. The HSR solves the problem of the rapid transportation of many passengers for domestic Chinese travel [1] and plays an essential role in promoting economic development by accelerating the flow of information, labor, and technology [2]. At the macro and regional levels, HSR openings have energized regional accessibility and regional economic growth [3,4], and improved the industrial agglomeration level of central cities [5]. At firm level, information flows brought by the opening of an HSR can reduce the risk of stock price collapse [6] and advance a company's innovation level [7]. Although studies have shown that HSR openings influence corporate behavior, the literature has not examined its impact on corporate fraud.

The fraud of listed companies can seriously damage market integrity and investors' interests, which is not conducive to sustainability, health, and the stable development of capital markets. Although the supervision of regulators has continuously strengthened in recent years, fraud events continue to occur and garnish substantial public interest. For sample, in 2018, Kangdexin, a Chinese A-share listed company, falsely boosted its operating revenue, operating costs, R and D expenses, and sales expenses, resulting in a false increase in profit of 2.436 billion yuan in the 2018 financial report, accounting for 711.29% of the total audited profit. In 2018, Kangmei Pharmaceutical Co., Ltd.(Beijing, China) falsely increased its monetary capital by 36.188 billion yuan, accounting for 45.96% of its total assets and 108.24% of its net assets using nonfinancial bookkeeping and false bookkeeping. According to data disclosed by China's Securities Regulatory Commission

(CSRC), 117 fraud cases of listed companies in China occurred in 2010, and by 2020, the incidence of fraud had increased to 1488. Detailed changes are shown in Figure 1.

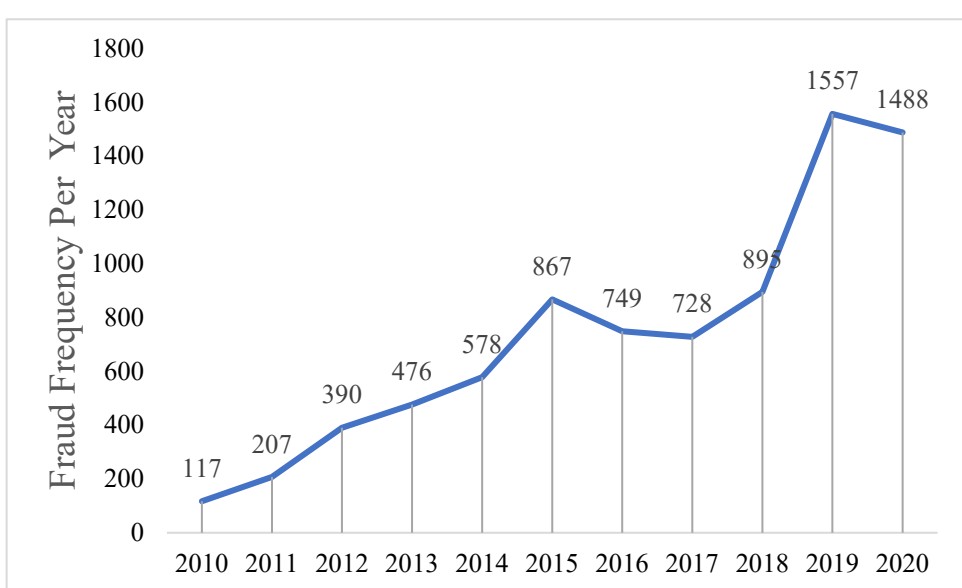

**Figure 1.** Annual fraud frequency of Chinese listed companies from 2015 to 2019.

Lowering corporate fraud, improving corporate governance, and protecting investors' legitimate rights and interests are important for sustainable, vibrant capital markets. Previous studies have considered that internal and external governance mechanisms such as executive characteristics [8], an Audit Committee [9], institutional investor supervision [10], and product market competition [11] are relevant factors determining corporate fraud. However, the role of transportation infrastructure in corporate fraud has not attracted scholars' attention. Does the development of transportation infrastructure affect corporate fraud? If so, what is the impact mechanism? This paper attempts to discuss the impact of HSR opening on corporate behavior.

Our paper selects China A-share listed companies from 2007 to 2020 and uses a difference-in-difference method to study the influence of HSR opening on corporate fraud and its mechanism. The empirical results show that HSR openings significantly affect corporate fraud by lowering the frequency of corporate fraud for firms in that city. Robustness tests, such as propensity score matching difference-in-difference tests (PSM-DID), placebo tests, introducing additional transportation infrastructure control variables, controlling for firm fixed effects, expanding the sample range, and eliminating the sample of large cities, support our premise that HSR introduction lowers corporate firm.

We demonstrate that HSR opening increases the external supervision of a company, reduces the financing constraints, and restricts fraud. When there is a low level of market competition and internal supervision, the effect of HSR opening on corporate fraud is more significant. After distinguishing the types of corporate fraud, we found that HSR openings mainly restrict information disclosure fraud and manager fraud but have no significant impact on operation fraud.

Our paper provides a new perspective for the research of corporate fraud by highlighting the impact of an exogenous event of improved transportation on firm behavior. Existing research on corporate fraud mainly focuses on external control and corporate governance, and few works of literature consider the impact of such objective traffic environment changes on corporate fraud. We focus on the effects of HSR on the microeconomic consequences and broaden the research perspectives of the economic consequences of HSR opening. The existing research on the impact of HSR large concentrates on the macroeconomy [2,3] and the regional economy [5]. Although several works discuss the impact of the opening of HSR on the risk of stock price collapse [6] and the level of corporate

innovation [7], fewer scholars have studied the relationship between HSR opening and corporate fraud. This paper provides new empirical evidence for the economic consequences of HSR by deepening our understanding of the relationship between transportation and corporate fraud and providing clues and theories for stakeholders to boost supervision, restrict corporate fraud, and promote sustainable corporate governance.

## 2. Literature Review

### 2.1. The Influence of Geographical Distance on Capital Market

Face-to-face communication leads to improved flows of "soft information" and provides an essential information source for investors in making investment decisions [12]. Geographical factors affect the acquisition of soft information and information asymmetry. Lerner found that the degree of venture capital participating in management in the company is related to distance. There is a positive correlation between the cost of information acquisition and the distance between the two parties; hence, a shortening of the geographical distance reduces the cost of information acquisition [13]. Coval and Moskowitz showed that mutual fund managers obtain higher returns through companies with close investment distances [14]. Additionally, studies show that geographic distance can reduce the degree of information asymmetry between stakeholders and companies, and impact resource allocation efficiency in capital markets [12,15]. Loughran compared central city firms, with more investors, and non-central city firms, with fewer investors nearby, finding that the closer the company is to its investors, the more accurate the internal information is transmitted [12].

Anderson and Wincoop demonstrate that the existence of geographical distance can cause information acquisition obstacles and information friction, thereby affecting the supervision costs of the parent company on the behavior of subsidiaries [16]. Malloy shows analysts closer to the tracked company made more accurate profit forecasts, and the geographical distance affects the accuracy of these forecasts [17].

### 2.2. The Influence of the HSR Opening on Economy

HSR plays a vital role in the rapid transportation of passenger flows between cities [1]. Most of HSR focuses on its economic impact. Chen believes that HSR accelerates the flow of the economy, information, labor, and technology [2]. Kim shows Korean HSR openings promote regional accessibility and economic growth [18]. Ahlfeldt and Feddersen conducted a study on the impact of opening the long Frankfurt HSR in Germany [3]. Their results document that HSR opening boosts the GDP of the cities along the line by 8.5%. Taking central China as a sample, Wang and Zhang found that HSR introduction strengthens commercial trade and communication flows between cities and regions and improves regional accessibility [19]. Lin et al. show the opening of HSR promotes the flow of personnel and funds between regions and thus affects the development of regional economies [20]. Some scholars focus on the relationship between HSR and industrial agglomeration and find that HSR openings improve a region's accessibility, enhance its market potential, and thus improve the industrial agglomeration level of the central city [5].

Academic works also examine HSR's impact on capital markets and firms. Vickerman found that HSR introduction influences corporate choice of location for its subsidiaries [21]. Willigers and Wee demonstrate that international companies prefer to choose a city with an international HSR as their office base [22]. Zhao et al. examine the influence of HSR on the capital market and its economic consequences from the perspective of stock price collapse risk by using the difference-in-difference method [6]. Chen et al. reveal that HSR openings boost information flows, mitigate information asymmetry inside and outside the firm, and positively impact executive compensation [23].

### 2.3. Factors Influencing Corporate Fraud

Most existing literature examines the influencing factors of company fraud from internal control and external control perspectives. Research has shown that internal controls

impact board features [24], senior management characteristics [8], audit committees [9], and the release of the corporate social responsibility report [25] on corporate fraud. External control impacts institutional supervision and analyst tracking [26], product market competition [11], and institutional investors [10] on corporate fraud.

Overall, the current academic research on HSR's impact on the economy mainly focuses on the macroeconomy and regional economy. However, a few papers examine the impact of HSR on firm behavior. This paper focuses on the importance of geography on corporate fraud.

## 3. Theoretical Analysis and Research Hypothesis

Current research on corporate fraud focuses on the fraud triangle theory [27–29]. The fraud triangle theory divides the main factors affecting corporate fraud into three aspects: incentive, opportunity, and rationalization. First, two aspects of incentives are generated, including capital market incentives and contract incentives. For example, manipulating stock prices to gain additional earnings [30], or easing debt pressures [31]. Second, "opportunity" usually refers to external or internal supervision, and when the level of internal and external supervision is high, the company may have fewer opportunities for fraud [32]. Third, the actual implementation of the fraud requires the fraudster to rationalize their behaviors from the moral level [27]. Therefore, some scholars find that manager characteristics are also an essential factor affecting corporate fraud [33].

We study whether the introduction of a HSR improves the supervision level of stakeholders and reduces fraud opportunities. Geographical distance can affect the supervision mechanism after the company's decision-making, which is not conducive to the stakeholders of the listed company to supervise the internal operation and decision-making of the listed company. Choi et al. found that distance affects the auditors' supervision ability [34]. When auditors are closer to listed companies, auditors can obtain more stakeholder evaluations of the company and better evaluate the managers' motivation and ability to manipulate earnings. The opening of an HSR breaks the constraint of geographical distance between stakeholders and listed companies, makes it more convenient for stakeholders to supervise listed companies, and reduces the opportunity of corporate fraud.

HSR openings can alleviate financing constraints faced by listed companies. Financial constraints and debt pressures are relevant reasons for corporate fraud [31]. With the opening of the HSR, the convenience of transportation between cities has increased, and the space–time distance between listed companies and capital providers is shortened. These changes increase the convenience for capital providers to conduct an on-site investigation and firm visit to the company, boost the ease to identify companies in trouble, reduce the risk of capital providers providing funds to listed companies, and alleviate capital pressure. Thus, increased HSR leads to increased transport accessibility and reduces the pressure, tendency, and frequency of corporate fraud.

**Hypothesis 1 (H1).** *HSR openings can inhibit corporate fraud.*

Further geographical distances increase the cost of communication between stakeholders and listed companies, produce corresponding moral hazards, and hinder the regular operation of the supervision mechanism after decision-making. For example, Kalnins and Lafontaine show that the agency cost and information asymmetry caused by larger geographical distances between the parent company and the subsidiary company significantly raise the supervision cost of the parent company leading to lower subsidiary performance [15]. Improved external supervision increases the probability of corporate fraud discovery and hence can inhibit corporate fraud. Based on the above analysis, we believe that external supervision plays a mediating role in the impact of HSR opening on corporate fraud and HSR openings themselves, thus increasing the ease of stakeholder supervision.

**Hypothesis 2 (H2).** *HSR openings can strengthen the external supervision of the company and then inhibit corporate fraud.*

Geographical distance is an essential factor affecting a firms financing costs. An increase of geographical distance between the company and the fund provider raises transaction costs, including information collection costs, communication costs, and supervision costs, which will impact the financing cost of the company. HSR openings reduce the time cost of personnel flow, help the capital provider thoroughly investigate the target company, reduce the investment risk of the fund provider, and alleviate financing constraints. Capital pressure is considered to be a critical inducing factor of corporate fraud [31]. Higher financing constraints trigger the motivation of corporate fraud and increase the tendency and frequency of corporate fraud. Therefore, financing constraints can play an intermediary role between improved transportation accessibility by HSR introduction and corporate fraud.

**Hypothesis 3 (H3).** *HSR openings can release the financing constraints of the company, and then inhibit corporate fraud.*

## 4. Research Design

### 4.1. Sample Data

This paper uses a difference-in-difference model to study the impact of HSRs on the fraud of A-share listed companies in China. We divided the sample into a treatment group and control as a function of whether HSR is near the location of the listed company. The sample period is 2007–2020. We also processed the data as follows: (1) we excluded financial listed firms and firms with missing data; (2) we dropped companies listed in the current year; and (3) all continuous variables were winsorized at the 1% and 99% levels [35]. The examined sample had 22,244 observations. HSR data was collected from the website of the State Railway Administration, and data on corporate fraud and other financial data of listed firms were found on the CSMAR database.

### 4.2. Variable Definition

#### 4.2.1. Corporate Fraud

We measured corporate fraud behavior in two ways:

- Fraud. If the company is disclosed to have committed fraud in the current year, it equals 1; otherwise, it equals 0.
- Frequency. The total frequency of fraud by the company in the current year disclosed by the CSRC.

#### 4.2.2. Other Control Variables

We used the following control variables: the size of the company (Size), the ratio of assets and liabilities (Lev), the growth rate of operating income (Growth), the value of Tobin Q (TobinQ), the equity nature of the company (Pattern), whether the auditor is from the big four accounting firms (Bigfour), degree of concentration of equity (Top10), the analyst followings (Anarpt), and the stock exchange rate (Turnover). The specific definitions and measurement methods of variables are listed in Table 1.

<p style="text-align: center;">**Table 1.** Variable definitions.</p>

| Variable Type | Variable Symbol | The Meaning of Variables and the Measurement Method |
|---|---|---|
| Explained variable | Fraud | Dumb variable, 1 for the corporate fraud of the disclosure in the current year, otherwise 0 |
| | Freq | Total number of frauds disclosed by the company in the current year |
| Explanatory variable | Train | If an HSR opening impacts the listed firm, it is included in the treatment group and equals 1; otherwise, it is the control group, and equals 0 |
| | TrainPost | The year of the listed company's office after the HSR opening is 1, otherwise it is 0 |
| Control variable | Size | Natural logarithm of a company's total assets |
| | Lev | Year-end total liabilities divide by total year-end assets |
| | Growth | The growth rate of the company's operating income in the current year |
| | TobinQ | Market value divides total assets in the current year. |
| | Pattern | For listed companies, value 1 for the state-owned firm, and 0 for the non-state-owned firm. |
| | Top10 | The shareholding ratio of top ten shareholders |
| | BigFour | Dummy variable, when the audit institution is the big four accounting firms value 1, otherwise value 0 |
| | Anarpt | Natural log of the number of analysts' reports in the current year |
| | Turnover | The turnover rate of the company's stock in the current year |

### 4.3. Model Design

We used a difference-in-difference model to test the impact of HSR introduction on the corporate fraud. The model of hypothesis 1 is as follows:

$$\text{Fraud}_{i,t} = \beta_0 + \beta_1 \text{Train}_{i,t} + \beta_2 \text{TrainPost}_{i,t} + \beta_3 \text{Control}_{i,t} + \text{Year}_{i,t} + \text{Ind}_{i,t} + \varepsilon_{i,t} \tag{1}$$

$$\text{Frequency}_{i,t} = \beta_0 + \beta_1 \text{Train}_{i,t} + \beta_2 \text{TrainPost}_{i,t} + \beta_3 \text{Control}_{i,t} + \text{Year}_{i,t} + \text{Ind}_{i,t} + \varepsilon_{i,t} \tag{2}$$

The dependent variable is our fraud dummy variable in (1), and we used a probit and logit regression to estimate the model. In (2), for frequency, we applied a Poisson regression and negative binomial regression to estimate the model. The explanatory variables are Train and Trainpost. When train = 1, the observation is the treatment group; when train = 0, their observation is coded as in the control group. When Trainpost = 1, the local HSR of the listed company is opened; when Trainpost = 0, HSR is not available in the city where the corporate headquarters are located. $\beta_2$ is the coefficient of interest. Our research hypothesis predicts $\beta_2 < 0$, implying that HSR introduction inhibits corporate fraud.

In order to further clarify the framework of this study, we supplemented the frame diagram, as shown in Figure 2. In the first part of the empirical results, the article highlights the regression results of Hypothesis 1. Next, we conduct a series of robustness tests to ensure that our conclusions are persuasive. Then, our paper tests the mediating effect of the two mechanisms proposed in Hypotheses 2 and 3. Finally, this paper further analyzes the results of the main hypothesis in different situations.

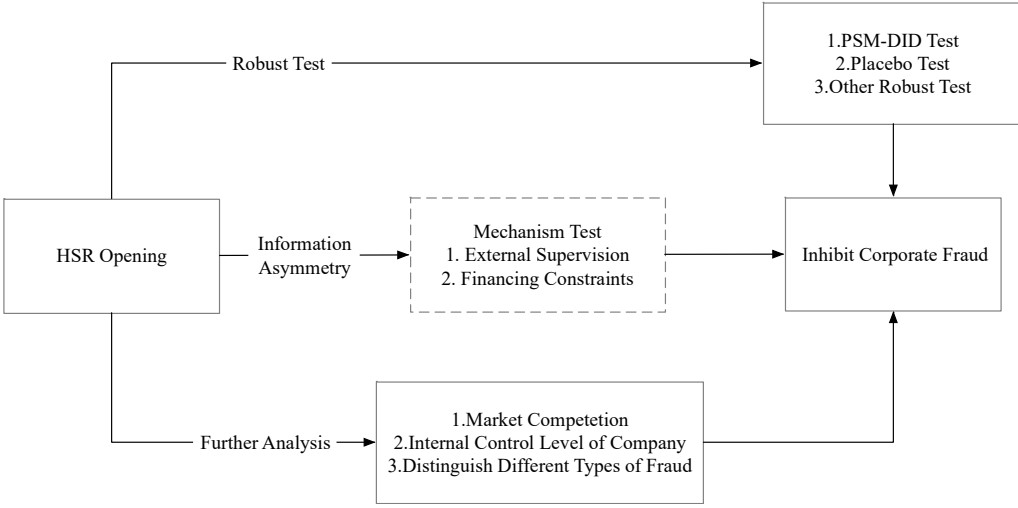

**Figure 2.** The research analysis framework.

## 5. Empirical Results and Analysis

### 5.1. Descriptive Statistical Results

The descriptive statistical results of the main variables are shown in Table 2. It can be seen that the mean values of fraud and freq are 0.1220 and 0.1762, respectively, indicating that the listed companies with fraud in the sample account for 12.20% of the sample. The maximum number of frauds is 38, which are the fraud events of Shaanxi Construction Machinery Co., Ltd. (Taiyuan, China) (600984) investigated and announced by CSRC in 2019. The variable's mean value of HSR introduction is 0.8427, indicating that approximately 84.27% of listed enterprises in the sample have experienced an HSR opening during the sample period.

**Table 2.** Descriptive statistical analysis.

| VarName | Obs | Mean | SD | Min | Median | Median |
|---|---|---|---|---|---|---|
| Fraud | 22,244 | 0.1220 | 0.3273 | 0.0000 | 0.0000 | 1.0000 |
| Freq | 22,244 | 0.1762 | 0.6573 | 0.0000 | 0.0000 | 38.0000 |
| Train | 22,244 | 0.8427 | 0.3641 | 0.0000 | 1.0000 | 1.0000 |
| TrainPost | 22,244 | 0.6760 | 0.4680 | 0.0000 | 1.0000 | 1.0000 |
| Size | 22,244 | 22.4246 | 1.4972 | 19.9263 | 22.1748 | 27.8520 |
| Lev | 22,244 | 0.4495 | 0.2125 | 0.0542 | 0.4446 | 0.9354 |
| Growth | 22,244 | 0.2125 | 0.4576 | −0.4984 | 0.1296 | 3.0733 |
| TobinQ | 22,244 | 2.0020 | 1.2423 | 0.8756 | 1.5875 | 8.0527 |
| Pattern | 22,244 | 0.4109 | 0.4920 | 0.0000 | 0.0000 | 1.0000 |
| Top10 | 22,244 | 0.5958 | 0.1511 | 0.2413 | 0.6051 | 0.9215 |
| BigFour | 22,244 | 1.9131 | 0.2818 | 1.0000 | 2.0000 | 2.0000 |
| Anarpt | 22,244 | 20.6655 | 24.3440 | 1.0000 | 11.0000 | 116.0000 |
| Turnover | 22,244 | 6.1032 | 0.8216 | 3.8132 | 6.1512 | 7.8238 |

### 5.2. Regression Results

This paper applies model (1) to the regression analysis of Hypothesis 1. Table 3 reports the results of H1. Columns (1) and (2) are the regression results of HSR on the corporate fraud tendency, which are estimated by probit regression and logit regression, respectively. Columns (3) and (4) are the regression results of the frequency of corporate fraud caused by HSR, which are estimated by Poisson regression and negative binomial regression, respectively. Columns (1) and (2) show the regression coefficients of Trainpost are −0.0928 and −0.1739, respectively, and are significant at the 10% level, indicating that the opening of HSR inhibits corporate fraud tendency. Regression results in columns (3) and (4) show the regression coefficients of Trainpost are −0.1468 and −0.1511, respectively,

which are both significant at the level of 10%, indicating that HSR openings also inhibit corporate frequency.

**Table 3.** HSR opening and company fraud.

| VarName | (1) Probit Fraud | (2) Logit Fraud | (3) Poisson Frep | (4) Nbreg Frep |
|---|---|---|---|---|
| Train | 0.0711 | 0.1385 * | 0.1048 | 0.0909 |
| | (1.6261) | (1.6702) | (1.5800) | (1.0722) |
| TrainPost | −0.0928 ** | −0.1739 ** | −0.1468 ** | −0.1511 ** |
| | (−2.4513) | (−2.4076) | (−2.5078) | (−2.0333) |
| Size | −0.0216 | −0.0394 | −0.0315 | −0.0339 |
| | (−1.3705) | (−1.3288) | (−1.4077) | (−1.1453) |
| Lev | 0.7295 *** | 1.3996 *** | 1.5009 *** | 1.4484 *** |
| | (10.2295) | (10.4903) | (14.8503) | (10.8886) |
| Growth | 0.0468 ** | 0.0808 * | 0.0812 ** | 0.0827 * |
| | (1.9844) | (1.8574) | (2.5503) | (1.9160) |
| TobinQ | 0.0293 ** | 0.0531 ** | 0.0600 *** | 0.0585 *** |
| | (2.5575) | (2.5005) | (3.7685) | (2.7410) |
| Pattern | −0.2188 *** | −0.4107 *** | −0.2620 *** | −0.3062 *** |
| | (−8.1320) | (−8.0536) | (−6.8095) | (−6.0876) |
| Top10 | −0.4340 *** | −0.8237 *** | −0.6101 *** | −0.6576 *** |
| | (−5.4469) | (−5.5616) | (−5.3845) | (−4.3410) |
| BigFour | 0.2092 *** | 0.4043 *** | 0.4849 *** | 0.4919 *** |
| | (4.0012) | (3.9268) | (6.1810) | (4.9692) |
| Anarpt | −0.0028 *** | −0.0055 *** | −0.0059 *** | −0.0059 *** |
| | (−4.8966) | (−4.9058) | (−6.7976) | (−5.3382) |
| Turnover | 0.0349 * | 0.0624 * | 0.0199 | 0.0372 |
| | (1.8884) | (1.8090) | (0.7676) | (1.0751) |
| Cons | −1.8429 *** | −3.3521 *** | −3.8665 *** | −3.8784 *** |
| | (−3.2532) | (−3.2332) | (−4.8706) | (−3.7012) |
| Year | Control | Control | Control | Control |
| Industry | Control | Control | Control | Control |
| N | 22244 | 22244 | 22244 | 22244 |
| PseudoR2 | 0.0506 | 0.0508 | 0.0695 | 0.0462 |

Notes: The *t*-values in parentheses, *** means the significance level is 1%, ** means the significance level is 5%, and * means the significance level is 10%.

In terms of control variables, the regression results in Table 3 show that the asset–liability ratio (Lev) has a significant positive correlation with the corporate fraud tendency and frequency, indicating that when the company's asset–liability level is higher, the corporate fraud tendency and frequency are higher, which is consistent with the results of Zhang's research [36]. The coefficient of company growth (Growth) is significantly positive, indicating that the faster the company grows, the more likely it is to violate rules. This may be because, in companies with higher growth, managers make more radical decisions and are more prone to fraud. This result is consistent with the research results of Cao et al. [37]. The sign of TobinQ is significantly positive because the larger TobinQ is, the more the firm increases investment expenditure, thereby raising the risk of fraud [37]. The coefficient of enterprise nature (Pattern) is significantly negative, indicating that the company has less tendency and number of violations in non-state-owned enterprises, which may be due to the more serious principal-agent problem in China's state-owned enterprises, and the coefficient is consistent with the results of existing literature [38]. The equity concentration (TOP10) is significantly negative, indicating that the companies with a higher equity concentration have a lower fraud tendency and frequency; this may be due to larger shareholders being more motivated to supervise the company when the equity concentration is higher. Bigfour's regression coefficient is significantly positive, indicating that when the company is audited by the four major accounting firms, the company has

more fraud tendency and frequency. We believe a possible explanation is that the audit quality of the four major accounting firms is higher; therefore, it is more likely to find and disclose corporate fraud. The coefficient of the number of analyst reports (Anarpt) is significantly negative; that is, the greater the number of analyst reports, the lower the corporate fraud tendency and frequency, indicating that analysts can play a good role in external supervision, and this result is consistent with the results of existing literature [37].

### 5.3. Robustness Test

5.3.1. PSM-DID Test

Our results may have a sample selection bias. This may occur because companies prone to fraud tend to establish headquarters in a city where a HSR has been opened. In order to avoid the possibility of selection bias, we applied a propensity score matching method to further test Hypothesis 1 [39]. The propensity score matching method can obtain control group samples that match the experimental group and delete the mismatched enterprises. Table 4 show the results of re-testing Hypothesis 1 for the matched samples. The results show that HSR (TrainPost) still has a significant negative correlation with corporate fraud (Fraud/Freq), indicating that the main test result is relatively robust.

**Table 4.** PSM-DID test.

| VarName | (1) Probit Fraud | (2) Logit Fraud | (3) Poisson Frep | (4) Nbreg Frep |
|---|---|---|---|---|
| Train | 0.0700 | 0.1374 | 0.1030 | 0.0901 |
| | (1.4735) | (1.5236) | (1.1318) | (1.0343) |
| TrainPost | −0.0951 ** | −0.1787 ** | −0.1500 ** | −0.1532 ** |
| | (−2.3912) | (−2.3644) | (−1.9903) | (−2.1251) |
| Size | −0.0387 ** | −0.0709 ** | −0.0521 | −0.0592 * |
| | (−2.4838) | (−2.4173) | (−1.5374) | (−1.8178) |
| Lev | 0.7367 *** | 1.4128 *** | 1.5115 *** | 1.4601 *** |
| | (9.6131) | (9.8120) | (9.9128) | (10.3058) |
| Growth | 0.0501 ** | 0.0860 * | 0.0861 * | 0.0913 ** |
| | (2.0190) | (1.8697) | (1.9280) | (2.1202) |
| TobinQ | 0.0230 * | 0.0416 * | 0.0536 ** | 0.0491 ** |
| | (1.8855) | (1.8304) | (2.2225) | (2.1390) |
| Pattern | −0.2255 *** | −0.4224 *** | −0.2689 *** | −0.3177 *** |
| | (−7.7854) | (−7.6439) | (−3.6702) | (−4.8925) |
| Top10 | −0.0044 *** | −0.0084 *** | −0.0062 *** | −0.0068 *** |
| | (−5.4719) | (−5.5562) | (−3.1154) | (−4.1081) |
| BigFour | 0.2094 *** | 0.4055 *** | 0.4791 *** | 0.4863 *** |
| | (3.7088) | (3.5706) | (2.8387) | (3.8458) |
| Anarpt | −0.0028 *** | −0.0054 *** | −0.0059 *** | −0.0058 *** |
| | (−4.5641) | (−4.5482) | (−3.8622) | (−4.2823) |
| Turnover | −0.0000 | −0.0001 | −0.0001 * | −0.0001 * |
| | (−1.4011) | (−1.4268) | (−1.7733) | (−1.7531) |
| Cons | −1.2381 ** | −2.2507 ** | −3.2460 *** | −3.0547 *** |
| | (−2.0587) | (−2.0638) | (−2.8220) | (−2.9684) |
| Year | Control | Control | Control | Control |
| Industry | Control | Control | Control | Control |
| N | 22220 | 22220 | 22220 | 22220 |
| PseudoR2 | 0.0504 | 0.0505 | | 0.0460 |

Notes: The *t*-values in parentheses, *** means the significance level is 1%, ** means the significance level is 5%, and * means the significance level is 10%.

5.3.2. Placebo Test

We applied a placebo test to avoid hypothesis 1 regression results driven by some accidental factors; we set up a virtual HSR opening time (TrainPostF1) one year before the opening of the HSR to conduct a trial. If there is a robust relationship between HSR

and corporate fraud, then we should not observe a significant change in corporate fraud before HSR opening. The results in Table 5 show that the coefficient of the dummy variable TrainPostF1 is not significant, which indicates that the main test results of HSR opening and corporate fraud are not caused by the time trend effect but by the causal effect of HSR openings.

**Table 5.** Placebo test.

| VarName | (1) Probit Fraud | (2) Logit Fraud | (3) Poisson Frep | (4) Nbreg Frep |
|---|---|---|---|---|
| TrainF1 | −0.0178 | −0.0357 | −0.0354 | −0.0473 |
| | (−0.4210) | (−0.4360) | (−0.5245) | (−0.5644) |
| TrainPostF1 | 0.0154 | 0.0330 | 0.0120 | 0.0246 |
| | (0.3695) | (0.4064) | (0.1790) | (0.2958) |
| Size | −0.0219 | −0.0394 | −0.0312 | −0.0347 |
| | (−1.3836) | (−1.3277) | (−1.3940) | (−1.1714) |
| Lev | 0.7342 *** | 1.4088 *** | 1.5100 *** | 1.4593 *** |
| | (10.3000) | (10.5656) | (14.9437) | (10.9715) |
| Growth | 0.0476 ** | 0.0829 * | 0.0826 *** | 0.0837 * |
| | (2.0170) | (1.9058) | (2.5954) | (1.9391) |
| TobinQ | 0.0291 ** | 0.0529 ** | 0.0602 *** | 0.0584 *** |
| | (2.5348) | (2.4902) | (3.7801) | (2.7356) |
| Pattern | −0.2191 *** | −0.4114 *** | −0.2621 *** | −0.3079 *** |
| | (−8.1469) | (−8.0648) | (−6.8159) | (−6.1302) |
| Top10 | −0.4420 *** | −0.8389 *** | −0.6214 *** | −0.6673 *** |
| | (−5.5492) | (−5.6644) | (−5.4828) | (−4.4050) |
| BigFour | 0.2107 *** | 0.4072 *** | 0.4871 *** | 0.4937 *** |
| | (4.0317) | (3.9557) | (6.2099) | (4.9886) |
| Anarpt | −0.0028 *** | −0.0055 *** | −0.0059 *** | −0.0059 *** |
| | (−4.9191) | (−4.9400) | (−6.8506) | (−5.3608) |
| Turnover | 0.0355 * | 0.0637 * | 0.0206 | 0.0376 |
| | (1.9195) | (1.8473) | (0.7925) | (1.0853) |
| Cons | −1.7931 *** | −3.2622 *** | −3.8007 *** | −3.8056 *** |
| | (−3.1693) | (−3.1506) | (−4.7965) | (−3.6342) |
| Year | Control | Control | Control | Control |
| Industry | Control | Control | Control | Control |
| N | 22244 | 22244 | 22244 | 22244 |
| PseudoR2 | 0.0504 | 0.0505 | 0.0691 | 0.0459 |

Notes: The *t*-values in parentheses, *** means the significance level is 1%, ** means the significance level is 5%, and * means the significance level is 10%.

### 5.3.3. Control the Impact of Other Transport Infrastructure

Although the difference-in-difference method adeptly controls the influence of other confounding factors, for robustness, we considered the influence of other transportation infrastructure. This helps further explain that it is the HSR opening rather than other changes in transportation infrastructure that lead to firm fraud. Therefore, following Cao and Zhang, we created a control for the natural logarithm of the annual passenger throughput of airports at the city level (Lnair) in the model [37]; data were collected from the China National Aviation Administration website. Table 6 show the regression results. The coefficient of HSR opening (Trainpost) is still negative with a significance level of 10%, indicating that after controlling the impact of other infrastructure, HSR can still have inhibited corporate fraud, and the results are relatively stable.

**Table 6.** Controlling the impact of other transport infrastructure.

| | (1) | (2) | (3) | (4) |
|---|---|---|---|---|
| | Probit | Logit | Poisson | Nbreg |
| Variable | Fraud | Fraud | Frep | Frep |
| Train | 0.0712 | 0.1376 * | 0.1053 | 0.0937 |
| | (1.6256) | (1.6573) | (1.5868) | (1.1037) |
| TrainPost | −0.0927 ** | −0.1751 ** | −0.1460 ** | −0.1480 ** |
| | (−2.4453) | (−2.4183) | (−2.4899) | (−1.9883) |
| LnAir | −0.0001 | 0.0012 | −0.0008 | −0.0039 |
| | (−0.0302) | (0.2420) | (−0.2002) | (−0.7694) |
| Size | −0.0216 | −0.0395 | −0.0314 | −0.0334 |
| | (−1.3699) | (−1.3328) | (−1.4029) | (−1.1293) |
| Lev | 0.7294 *** | 1.4004 *** | 1.5004 *** | 1.4459 *** |
| | (10.2248) | (10.4928) | (14.8424) | (10.8663) |
| Growth | 0.0468 ** | 0.0806 * | 0.0813 ** | 0.0830 * |
| | (1.9846) | (1.8534) | (2.5539) | (1.9217) |
| TobinQ | 0.0293** | 0.0529** | 0.0601 *** | 0.0592 *** |
| | (2.5575) | (2.4930) | (3.7734) | (2.7713) |
| Pattern | −0.2188 *** | −0.4110 *** | −0.2618 *** | −0.3051 *** |
| | (−8.1276) | (−8.0569) | (−6.8026) | (−6.0609) |
| Top10 | −0.4339 *** | −0.8244 *** | −0.6096 *** | −0.6554 *** |
| | (−5.4451) | (−5.5655) | (−5.3797) | (−4.3257) |
| BigFour | 0.2092 *** | 0.4050 *** | 0.4844 *** | 0.4903 *** |
| | (3.9989) | (3.9321) | (6.1727) | (4.9521) |
| Anarpt | −0.0028 *** | −0.0055 *** | −0.0059 *** | −0.0059 *** |
| | (−4.8966) | (−4.9059) | (−6.7979) | (−5.3456) |
| Turnover | 0.0349 * | 0.0624 * | 0.0199 | 0.0376 |
| | (1.8885) | (1.8084) | (0.7676) | (1.0866) |
| Cons | −1.8429 *** | −3.3513 *** | −3.8674 *** | −3.8872 *** |
| | (−3.2533) | (−3.2324) | (−4.8715) | (−3.7091) |
| Year | Control | Control | Control | Control |
| Industry | Control | Control | Control | Control |
| N | 22244 | 22244 | 22244 | 22244 |
| PseudoR2 | 0.0507 | 0.0509 | 0.0693 | 0.0462 |

Notes: The *t*-values in parentheses, *** means the significance level is 1%, ** means the significance level is 5%, and * means the significance level is 10%.

### 5.3.4. Firm Fixed Effect

Although the regression model in this paper controls many influencing factors, there may still be the problem of missing variables. The company fixed-effect model can better avoid the result deviation caused by the existence of missing variables. Following Meng et al. [39], we applied firm fixed effects and retest Hypothesis 1. Note, after controlling the fixed effect model at the company level, the variable Train was excluded due to multicollinearity. The results are shown in Table 7. Xtlogit and Xtpoisson are the fixed-effect logit model and fixed-effect Poisson regression model, respectively. According to the regression results in Table 7, after controlling the fixed effect at the company level, the coefficient of HSR was still significantly negative. The above results are consistent with the previous results, which improves the robustness of our conclusion.

**Table 7.** Company fixed effect model.

| Variable | (1) XtLogit Fraud | (2) Xtlogit Fraud | (3) XtPoisson Freq | (4) XtPoisson Freq |
|---|---|---|---|---|
| TrainPost | −0.2518 ** | −0.2449 ** | −0.1495 * | −0.1401 * |
| | (−2.4164) | (−2.3444) | (−1.8033) | (−1.6886) |
| Size | | −0.1911 *** | | −0.1170 ** |
| | | (−2.7745) | | (−2.1565) |
| Lev | | 1.2393 *** | | 1.2754 *** |
| | | (5.1163) | | (6.9309) |
| Growth | | −0.0121 | | −0.0464 |
| | | (−0.2461) | | (−1.2944) |
| TobinQ | | 0.0082 | | 0.0260 |
| | | (0.2678) | | (1.1039) |
| Pattern | | −0.1387 | | −0.1184 |
| | | (−0.7955) | | (−0.8835) |
| Top10 | | −0.1403 | | −0.0114 |
| | | (−0.4473) | | (−0.0471) |
| BigFour | | 0.1863 | | −0.2482 |
| | | (0.8338) | | (−1.5616) |
| Anarpt | | −0.0016 | | −0.0007 |
| | | (−0.9604) | | (−0.5655) |
| Turnover | | 0.1418 *** | | 0.1438 *** |
| | | (2.9556) | | (4.0374) |
| Year | Control | Control | Control | Control |
| Industry | Control | Control | Control | Control |
| N | 13044 | 13044 | 13078 | 13078 |

Notes: The *t*-values in parentheses, *** means the significance level is 1%, ** means the significance level is 5%, and * means the significance level is 10%.

### 5.3.5. Expand the Sample Range

Another factor that may affect the regression results of the main test is the sample interval. To solve this potential problem, we expanded the sample interval from 2000 to 2020 because the first HSR in China was experimentally operated in 2003 [6]. The results in Table 8 show HSR can still curb the corporate fraud tendency and frequency significantly, which indicates that our research results are not affected by the sample interval and have strong robustness.

### 5.3.6. Eliminate the Influence of Big Cities

When planning the HSR, the government may consider the role of the central city and give more consideration to the construction of HSR in the central city. This policy preference may also cause endogenous problems, which affect the accuracy of our regression results to a certain extent. Therefore, referring to previous studies [40], we excluded Beijing, Shanghai, Guangzhou, Zhengzhou, Chengdu, Wuhan, Xi'an, Shenyang, and other cities with higher GDP ranking, and retaining a total of 16,196 samples. The empirical results are shown in Table 9. The results indicate that the coefficient of HSR is still significantly negative, which verifies the robustness of our hypothesis.

**Table 8.** Expanded sample range.

| Variable | (1) Probit Fraud | (2) Logit Fraud | (3) Poisson Frep | (4) Nbreg Frep |
|---|---|---|---|---|
| Train | 0.0660 | 0.1418 * | 0.1128 * | 0.0887 |
| | (1.5676) | (1.7488) | (1.7256) | (1.0611) |
| TrainPost | −0.0921 ** | −0.1825 ** | −0.1655 *** | −0.1504 ** |
| | (−2.4870) | (−2.5597) | (−2.8451) | (−2.0356) |
| Size | −0.0193 | −0.0391 | 0.0143 | −0.0515 ** |
| | (−1.4749) | (−1.5343) | (0.8587) | (−2.0234) |
| Lev | 0.1752 *** | 0.4460 *** | 0.1390 *** | 0.9226 *** |
| | (4.8960) | (4.6845) | (6.4369) | (7.5651) |
| Growth | −0.0001 | −0.0002 | −0.0003 | −0.0004 |
| | (−0.2457) | (−0.2526) | (−0.2799) | (−0.2756) |
| TobinQ | −0.0170 *** | −0.0314 ** | −0.0127 *** | 0.0006 |
| | (−3.4477) | (−2.2080) | (−3.4668) | (0.0726) |
| Pattern | −0.2117 *** | −0.4030 *** | −0.2411 *** | −0.3123 *** |
| | (−8.1455) | (−8.0732) | (−6.3562) | (−6.2834) |
| Top10 | −0.0056 *** | −0.0103 *** | −0.0086 *** | −0.0079 *** |
| | (−7.4007) | (−7.2119) | (−7.9361) | (−5.3207) |
| BigFour | 0.2207 *** | 0.4224 *** | 0.4914 *** | 0.4996 *** |
| | (4.3023) | (4.1422) | (6.2848) | (5.0552) |
| Anarpt | −0.0025 *** | −0.0049 *** | −0.0062 *** | −0.0052 *** |
| | (−4.8012) | (−4.7343) | (−7.7342) | (−5.0553) |
| Turnover | −0.0000 | −0.0001 | −0.0001** | −0.0001* |
| | (−1.5666) | (−1.5611) | (−2.3367) | (−1.8060) |
| Cons | −1.2488 ** | −2.2616 ** | −3.6522 *** | −2.8109 *** |
| | (−2.4887) | (−2.4105) | (−5.1806) | (−2.9696) |
| Year | Control | Control | Control | Control |
| Industry | Control | Control | Control | Control |
| N | 25088 | 25088 | 25088 | 25088 |
| PseudoR2 | 0.0653 | 0.0652 | 0.0800 | 0.0586 |

Notes: The *t*-values in parentheses, *** means the significance level is 1%, ** means the significance level is 5%, and * means the significance level is 10%.

**Table 9.** Excluding the impact of big cities.

| Variable | (1) Probit Fraud | (2) Logit Fraud | (3) Poisson Frep | (4) Nbreg Frep |
|---|---|---|---|---|
| Train | 0.0983 ** | 0.1877 ** | 0.1574 ** | 0.1285 |
| | (2.0903) | (2.1120) | (2.2185) | (1.4397) |
| TrainPost | −0.0931 ** | −0.1743 ** | −0.1590 ** | −0.1651 ** |
| | (−2.2297) | (−2.2052) | (−2.5050) | (−2.0682) |
| Size | −0.0064 | −0.0119 | 0.0191 | 0.0212 |
| | (−0.3423) | (−0.3431) | (0.7325) | (0.6180) |
| Lev | 0.6775 *** | 1.2957 *** | 1.3525 *** | 1.2885 *** |
| | (8.2269) | (8.4423) | (11.6226) | (8.6086) |
| Growth | 0.0453 | 0.0801 | 0.0774 ** | 0.0671 |
| | (1.6359) | (1.5897) | (2.1172) | (1.3611) |
| TobinQ | 0.0338 ** | 0.0615 ** | 0.0840 *** | 0.0764 *** |
| | (2.5191) | (2.4988) | (4.5937) | (3.1257) |
| Pattern | −0.1975 *** | −0.3667 *** | −0.2225 *** | −0.2791 *** |
| | (−6.3437) | (−6.2787) | (−5.0621) | (−4.8267) |
| Top10 | −0.3361 *** | −0.6417 *** | −0.3565 *** | −0.4615 *** |
| | (−3.6660) | (−3.7897) | (−2.7532) | (−2.7035) |
| BigFour | 0.1717 *** | 0.3314 *** | 0.4177 *** | 0.4202 *** |
| | (2.6046) | (2.6040) | (4.3613) | (3.4147) |

| Variable | (1) | (2) | (3) | (4) |
|---|---|---|---|---|
| | Probit | Logit | Poisson | Nbreg |
| | Fraud | Fraud | Frep | Frep |
| Anarpt | −0.0032 *** | −0.0062 *** | −0.0083 *** | −0.0078 *** |
| | (−4.7760) | (−4.7753) | (−8.0324) | (−6.0827) |
| Turnover | 0.0169 | 0.0268 | −0.0069 | 0.0233 |
| | (0.7845) | (0.6735) | (−0.2319) | (0.5959) |
| Cons | −0.5518 | −1.3013 | −3.5016 *** | −3.4924 *** |
| | (−0.6424) | (−0.8702) | (−4.0426) | (−2.5965) |
| Year | Control | Control | Control | Control |
| Industry | Control | Control | Control | Control |
| N | 16196 | 16196 | 16196 | 16196 |
| PseudoR2 | 0.0447 | 0.0451 | 0.0663 | 0.0434 |

Notes: The *t*-values in parentheses, *** means the significance level is 1%, ** means the significance level is 5%, and * means the significance level is 10%.

## 6. Analysis of the Impact Mechanism

In the previous research hypothesis, HSR can reduce the cost of stakeholder supervision and ease financing constraints to curb corporate fraud. Next, we will further examine the intermediary effect of external supervision and financing constraints.

### 6.1. External Supervision

We used model (3) and model (4) to test H2. We used the number of on-site investigations of listed companies by investors to indicate the level of supervision of corporate stakeholders. The more on-site investigations conducted by investors, the higher the level of external supervision of the company.

$$\text{Research}_{i,t} = \beta_0 + \beta_1 \text{Train}_{i,t} + \beta_2 \text{TrainPost}_{i,t} + \beta_3 \text{Control}_{i,t} + \text{Year}_{i,t} + \text{Ind}_{i,t} + \varepsilon_{i,t} \tag{3}$$

$$\text{Depvar}_{i,t} = \beta_0 + \beta_1 \text{Research}_{i,t} + \beta_2 \text{Train}_{i,t} + \beta_3 \text{TrainPost}_{i,t} + \beta_4 \text{Control}_{i,t} + \text{Year}_{i,t} + \text{Ind}_{i,t} + \varepsilon_{i,t} \tag{4}$$

Table 10 (1) show the regression results of model (3). The coefficient of HSR openings is 0.5214 and significant at 1%, indicating that HSR openings increase the number of field investigations by investors. The regression results of model (4) are listed in column (2) and column (3), showing that the more field research by external investors, the lower the company's tendency and frequency of fraud, indicating that external supervision is an important path for the influence of HSR on corporate fraud. This is consistent with the expectation of hypothesis 2.

### 6.2. Financing Constraints

We used models (4) and (5) to test H3. Referring to Kaplan and Zingales [41], we used the KZ index to measure financing constraints. Specifically, we constructed the KZ index according to the steps below:

(1) For each year of the entire sample, we collected and calculated the following data: the operating net cash flow divided by total assets of the previous period ($\text{CF}_{it}/\text{A}_{it-1}$), cash dividends divided by total assets of the previous period ($\text{DIV}_{it}/\text{A}_{it-1}$), cash holdings divided by total assets of the previous period ($\text{C}_{it}/\text{A}_{it-1}$), asset-liability ratio ($\text{LEV}_{it}$) and Tobin'sQ ($\text{TobinQ}_{it}$). If $\text{CF}_{it}/\text{A}_{it-1}$ is lower than the median, then kz1 equals 1, otherwise, it equals 0; if $\text{DIV}_{it}/\text{A}_{it-1}$ is lower than the median, kz2 equals 1, otherwise, it equals 0; if $\text{C}_{it}/\text{A}_{it-1}$ is lower than the median, kz3 equals 1, otherwise, kz3 equals 0; if $\text{LEV}_{it}$ is higher than the median, kz4 equals 1, otherwise, kz4 equals 0; if $\text{TobinQ}_{it}$ is higher than the median, kz5 equals 1, otherwise, kz5 equals 0.

(2) Calculating the KZ index. KZ = kz1 + kz2 + kz3 + kz4 + kz5.

(3)   We took the KZ index as the dependent variable to regress $CF_{it}/A_{it-1}$, $DIV_{it}/A_{it-1}$, $C_{it}/A_{it-1}$, $LEV_{it}$, and $TobinQ_{it}$ and estimate the regression of each variable coefficient.

**Table 10.** External supervision, HSR opening, and corporate fraud.

| Variable | (1) | (2) | (3) |
|---|---|---|---|
| | Research | Fraud | Freq |
| Train | 0.4855 *** | 0.0735 * | 0.1063 |
| | (3.6139) | (1.6802) | (1.6035) |
| TrainPost | 0.5214 *** | −0.0896 ** | −0.1431 ** |
| | (4.5716) | (−2.3652) | (−2.4439) |
| Research | | −0.0057 ** | −0.0062 * |
| | | (−2.5396) | (−1.8735) |
| Size | −0.0186 | −0.0219 | −0.0317 |
| | (−0.3782) | (−1.3894) | (−1.4187) |
| Lev | −0.6090 *** | 0.7262 *** | 1.4942 *** |
| | (−2.6741) | (10.1802) | (14.7815) |
| Growth | 0.1525 * | 0.0475 ** | 0.0819 *** |
| | (1.9298) | (2.0144) | (2.5759) |
| TobinQ | −0.0191 | 0.0291 ** | 0.0596 *** |
| | (−0.5086) | (2.5372) | (3.7468) |
| Pattern | −0.9702 *** | −0.2241 *** | −0.2681 *** |
| | (−11.4968) | (−8.3043) | (−6.9468) |
| Top10 | −0.4525 * | −0.4340 *** | −0.6089 *** |
| | (−1.8005) | (−5.4473) | (−5.3772) |
| BigFour | −0.4723 *** | 0.2069 *** | 0.4825 *** |
| | (−3.1950) | (3.9559) | (6.1480) |
| Anarpt | 0.0478 *** | −0.0025 *** | −0.0056 *** |
| | (27.1543) | (−4.2953) | (−6.3017) |
| Turnover | 0.5005 *** | 0.0380 ** | 0.0227 |
| | (8.5717) | (2.0478) | (0.8735) |
| Cons | −4.5375 ** | −1.8670 *** | −3.8872 *** |
| | (−2.4338) | (−3.2960) | (−4.8969) |
| Year | Control | Control | Control |
| Industry | Control | Control | Control |
| N | 22244 | 22244 | 22244 |
| AdjR2/PseudoR2 | 0.1401 | 0.0512 | 0.0695 |

Notes: The *t*-values in parentheses, *** means the significance level is 1%, ** means the significance level is 5%, and * means the significance level is 10%.

Using the results of the logistic regression model, we can calculate the KZ index of the degree of financing constraints of each listed company. The larger the KZ index, the higher the degree of financing constraints faced by the listed company. We report the empirical results in Table 3.

$$KZ_{i,t} = \beta_0 + \beta_1 Train_{i,t} + \beta_2 TrainPost_{i,t} + \beta_3 Control_{i,t} + Year_{i,t} + Ind_{i,t} + \varepsilon_{i,t} \tag{5}$$

$$Depvar_{i,t} = \beta_0 + \beta_1 KZ_{i,t} + \beta_2 Train_{i,t} + \beta_3 TrainPost_{i,t} + \beta_4 Control_{i,t} + Year_{i,t} + Ind_{i,t} + \varepsilon_{i,t} \tag{6}$$

Column (1) of Table 11 shows the regression results of model (5), in which the coefficient of HSR is significantly negative at the level of 1%. The regression results of model (6) are listed in columns (2) and (3). The results show that the higher the degree of financing constraints, the higher the tendency and frequency of corporate fraud. The results show that HSR affects corporate fraud by alleviating financing constraints. This is consistent with the expectation of hypothesis 3.

**Table 11.** Financing constraints, HSR opening, and corporate fraud.

| Variable | (1) | (2) | (3) |
| --- | --- | --- | --- |
| | **KZ** | **Fraud** | **Freq** |
| Train | −0.0077 | 0.0815 * | 0.1064 |
| | (−0.2001) | (1.7338) | (1.5160) |
| TrainPost | −0.0562 * | −0.0945 ** | −0.1529 ** |
| | (−1.7453) | (−2.3465) | (−2.4844) |
| KZ | | 0.0150 * | 0.0255 * |
| | | (1.6982) | (1.9370) |
| Size | −0.0512 *** | −0.0221 | −0.0197 |
| | (−3.7355) | (−1.2864) | (−0.8023) |
| Lev | 7.9829 | 0.6683 *** | 1.3989 *** |
| | (127.8542) | (6.4074) | (9.4013) |
| Growth | −0.9714 | 0.0386 | 0.0548 |
| | (−43.4055) | (1.4458) | (1.4757) |
| TobinQ | −0.0481 *** | 0.0291 ** | 0.0589 *** |
| | (−4.5407) | (2.3502) | (3.4414) |
| Pattern | −0.0696 *** | −0.2449 *** | −0.3555 *** |
| | (−2.9822) | (−8.4704) | (−8.4025) |
| Top10 | −1.1561 *** | −0.2492 *** | −0.4156 *** |
| | (−15.4604) | (−2.7498) | (−3.2018) |
| BigFour | −0.0649 | 0.1733 *** | 0.4906 *** |
| | (−1.4994) | (2.8998) | (4.8730) |
| Anarpt | −0.0088 *** | −0.0026 *** | −0.0055 *** |
| | (−17.6518) | (−4.0990) | (−5.7188) |
| Turnover | −0.0087 | 0.1038 *** | 0.1561 *** |
| | (−0.4866) | (4.7993) | (5.0756) |
| Cons | −0.3381 | −2.2478 *** | −4.9172 *** |
| | (−0.8859) | (−3.7298) | (−5.8083) |
| Year | Control | Control | Control |
| Industry | Control | Control | Control |
| N | 18727 | 18726 | 18727 |
| AdjR2/PseudoR2 | 0.5947 | 0.0549 | 0.0686 |

Notes: The *t*-values in parentheses, *** means the significance level is 1%, ** means the significance level is 5%, and * means the significance level is 10%.

## 7. Further Analysis

### 7.1. Market Competition

HSR can accelerate the flow of population and information between regions and improve the company's information transparency. According to existing studies, market competition exerts external pressure on a firm's information disclosure and then promotes the improvement of the company's information disclosure quality [11]. This means that when the market competition is high, the company's information disclosure quality is high, and the company has fewer opportunities for fraud. When this occurs, the inhibition effect of HSR on corporate fraud is limited. When the degree of market competition is low, the quality of company information disclosure is also low. When this occurs, HSR openings more effectively inhibit company fraud. Thus, we further investigate the regulatory effect of market competition on the relationship between opening an HSR and corporate fraud.

HHI is usually used to measure product market competition [11,42]. The smaller the HHI, the more competitive the product market is. According to the HHI of the company industry, we divided the companies into two groups: high market competition and low market competition. Table 12 show the regression results. In the low market competition groups (1) (2), the HSR opening regression coefficients are −0.1170 and −0.2637, significant at 5% and 1%, respectively. The HSR opening regression coefficients are not significant in the low external supervision level groups (3) (4). The results show that market competition is an essential regulatory mechanism between HSRs opening and corporate fraud. When

market competition is low, opening an HSR has a more significant inhibitory effect on corporate fraud.

**Table 12.** Market competition, HSR opening, and corporate fraud.

| Variable | (1) | (2) | (3) | (4) |
|---|---|---|---|---|
| | **Low Market Competition** | | **High Market Competition** | |
| | **Fraud** | **Freq** | **Fraud** | **Freq** |
| Train | 0.1131 * | 0.1902 ** | 0.0293 | 0.0288 |
| | (1.8772) | (2.0951) | (0.4700) | (0.2988) |
| TrainPost | −0.1170 ** | −0.2637 *** | −0.0797 | −0.0884 |
| | (−2.1804) | (−3.2279) | (−1.5080) | (−1.0526) |
| Size | −0.0076 | −0.0185 | −0.0011 | 0.0645 ** |
| | (−0.4177) | (−0.7570) | (−0.0530) | (2.3468) |
| Lev | 0.3601 *** | 0.4904 *** | 0.1233 *** | 0.1018 *** |
| | (5.1935) | (9.3167) | (2.8653) | (2.6907) |
| Growth | −0.0000 | −0.0000 | −0.0024 | −0.0044 |
| | (−0.1552) | (−0.1230) | (−0.5659) | (−0.5772) |
| TobinQ | −0.0037 | −0.0198 | −0.0115 * | −0.0067 |
| | (−0.5728) | (−1.5248) | (−1.8873) | (−1.1548) |
| Pattern | −0.1968 *** | −0.1603 *** | −0.2210 *** | −0.3303 *** |
| | (−5.4764) | (−3.1712) | (−5.5880) | (−5.5369) |
| Top10 | −0.4968 *** | −0.4786 *** | −0.5658 *** | −1.1652 *** |
| | (−4.5638) | (−3.1298) | (−5.1224) | (−7.2194) |
| BigFour | 0.2419 *** | 0.4802 *** | 0.1634 * | 0.4184 *** |
| | (3.6561) | (5.0252) | (1.9119) | (2.9677) |
| Anarpt | −0.0028 *** | −0.0073 *** | −0.0022 *** | −0.0042 *** |
| | (−3.9481) | (−6.5002) | (−2.6656) | (−3.5246) |
| Turnover | 0.0298 | −0.0176 | 0.0192 | 0.0422 |
| | (1.2333) | (−0.5244) | (0.7319) | (1.0992) |
| Cons | −2.3585 *** | −4.4030 *** | −1.8640 *** | −5.2646 *** |
| | (−3.6344) | (−4.6138) | (−2.7760) | (−5.2431) |
| Year | Control | Control | Control | Control |
| Industry | Control | Control | Control | Control |
| N | 11803 | 11803 | 11210 | 11210 |
| PseudoR2 | 0.0618 | 0.0820 | 0.0441 | 0.0574 |

Notes: The *t*-values in parentheses, *** means the significance level is 1%, ** means the significance level is 5%, and * means the significance level is 10%.

### 7.2. Internal Control Level of the Company

The HSR opening can alleviate the information asymmetry inside and outside the company and reduce the company's opportunities for fraud. High-quality internal control can inhibit the company's earnings management level and prevent the actual occurrence of corporate misconduct [43]. Therefore, when the company's internal control quality is high, the company's internal management and operation are more standardized, and the company has fewer opportunities for fraud. At this time, the inhibition effect of HSR on corporate fraud is limited. When the company's internal control quality is low, the company has more opportunities and motivation to implement frauds. At this time, HSR can more effectively corrupt corporate fraud.

DiBo, the internal control index, is usually used to measure the quality of internal control [44]. The higher the DiBo internal control index, the better the internal control quality. Based on the average value of the DiBo internal control index in the industry in the year, we divided the companies into two groups, including a high internal control quality group and a low internal control quality group, and then regress the model (1), respectively. The regression results are shown in Table 13. In the low internal control level group, the HSR opening regression coefficients are −0.0995 and −0.1671, significant at 10% and 5%, respectively. In the high internal control level group, the HSR opening regression coefficients are not significant. The results indicate that the correlation between the opening

of HSR and corporate fraud is affected by the quality of internal control. If the internal control quality is low, opening an HSR can inhibit corporate fraud more significantly.

**Table 13.** Internal control, HSR opening, and corporate fraud.

| Variable | (1) | (2) | (3) | (4) |
|---|---|---|---|---|
| | **Low Internal Control Level** | | **High Internal Control Level** | |
| | **Fraud** | **Freq** | **Fraud** | **Freq** |
| Train | 0.1140 * | 0.2093 ** | 0.0422 | −0.0397 |
| | (1.8860) | (2.5018) | (0.6453) | (−0.3632) |
| TrainPost | −0.0995 * | −0.1671 ** | −0.0867 | −0.1469 |
| | (−1.8698) | (−2.2682) | (−1.5598) | (−1.5161) |
| Size | 0.0264 | 0.1117 *** | 0.0093 | 0.0429 |
| | (1.2773) | (4.7374) | (0.4602) | (1.2910) |
| Lev | 0.5406 *** | 0.0993 *** | 0.0981 ** | 0.2235 ** |
| | (6.0787) | (4.4825) | (2.0780) | (2.2610) |
| Growth | −0.0001 | −0.0002 | −0.0001 | −0.0001 |
| | (−0.3619) | (−0.3177) | (−0.0709) | (−0.0699) |
| TobinQ | 0.0148 | 0.0306 *** | −0.0078 | −0.0221 |
| | (1.5715) | (2.9902) | (−1.2012) | (−1.5597) |
| Pattern | −0.2294 *** | −0.1149 ** | −0.1680 *** | −0.3854 *** |
| | (−6.2201) | (−2.4092) | (−4.0641) | (−5.7567) |
| Top10 | −0.3120 *** | −0.4167 *** | −0.4846 *** | −1.0322 *** |
| | (−2.8249) | (−2.9294) | (−4.1344) | (−5.5978) |
| BigFour | 0.3611 *** | 0.8202 *** | 0.0846 | 0.1552 |
| | (4.3721) | (7.0773) | (1.1426) | (1.2962) |
| Anarpt | −0.0002 | −0.0030 *** | −0.0026 *** | −0.0065 *** |
| | (−0.2167) | (−2.6469) | (−3.4109) | (−4.9306) |
| Turnover | 0.0656 ** | 0.0632 * | 0.0207 | 0.0240 |
| | (2.5250) | (1.9082) | (0.7804) | (0.5652) |
| Cons | −2.8562 *** | −6.9188 *** | −2.5079 *** | −4.8601 *** |
| | (−3.4477) | (−6.6066) | (−3.1131) | (−3.8279) |
| Year | Control | Control | Control | Control |
| Industry | Control | Control | Control | Control |
| N | 9368 | 9376 | 12969 | 12969 |
| PseudoR2 | 0.0345 | 0.0436 | 0.0699 | 0.0920 |

Notes: The *t*-values in parentheses, *** means the significance level is 1%, ** means the significance level is 5%, and * means the significance level is 10%.

### 7.3. Distinguish Different Types of Fraud

In this section we refer to the classification of the types of corporate fraud by the CSRC, dividing the corporate fraud into information disclosure fraud, operation fraud, and manager fraud, further investigating the relationship between HSR and different types of corporate fraud. Information disclosure fraud includes fictitious profits, false records (misleading statements), and significant omissions. Operation fraud indicates fraudulent listing, illegal investment, unauthorized changes in funds, illegal guarantees, and improper general accounting treatments. Manager fraud includes insider trading, illegal buying and selling of stocks, manipulating stock prices, and embezzling company assets. The cumulative frequency of corporate fraud in the same year is shown in Table 14.

**Table 14.** Classified statistics of corporate fraud.

| Year | Information Disclosure Fraud | Operation Fraud | Manager Fraud |
|---|---|---|---|
| 2007 | 57 | 5 | 22 |
| 2008 | 38 | 5 | 43 |
| 2009 | 47 | 15 | 74 |
| 2010 | 50 | 9 | 49 |
| 2011 | 74 | 47 | 68 |
| 2012 | 170 | 58 | 87 |
| 2013 | 201 | 81 | 124 |
| 2014 | 241 | 71 | 136 |
| 2015 | 386 | 92 | 243 |
| 2016 | 382 | 77 | 171 |
| 2017 | 359 | 72 | 162 |
| 2018 | 447 | 64 | 164 |
| 2019 | 662 | 141 | 409 |
| 2020 | 642 | 168 | 430 |

Table 15 report the impact of HSR openings on different types of fraud. Columns (1) and (2) show the regression results of HSR introduction on information disclosure fraud tendency and frequency, and columns (3) and (4) show the regression results of HSR opening on operation fraud tendency and fraud frequency, columns (5) and (6) are the regression results of the HSR opening on managers' fraud tendency and fraud frequency. The results show that the impact of HSR on information disclosure fraud and manager fraud is still significant, but the impact on operation fraud is not significant. We think that a more reasonable explanation is that HSR provides more channels for information disclosure fraud and manager fraud, including easing information asymmetry, strengthening external supervision, and easing financing constraints. However, the impact of these three channels on the company's operating fraud is relatively limited. The improvement of information transparency and the strengthening of external supervision make stakeholders better understand the company's financial activities and accounting information. The easing of financing constraints reduces the cash flow pressure of managers, but these channels cannot directly affect the company's operating activities.

**Table 15.** HSR opening and corporate fraud: distinguishing types of fraud.

| | (1) | (2) | (3) | (4) | (5) | (6) |
|---|---|---|---|---|---|---|
| | Information Disclosure Fraud | | Corporate Fraud | | Manager Fraud | |
| Variable | Freq | Fraud | Freq | Fraud | Freq | Fraud |
| Train | 0.0125 | 0.0607 | 0.0288 | 0.0940 | 0.1129 * | 0.2317 ** |
| | (0.2130) | (0.5641) | (0.3636) | (0.5158) | (1.9305) | (1.9600) |
| TrainPost | −0.0945 * | −0.1812 * | −0.0706 | −0.1936 | −0.1220 ** | −0.2053 ** |
| | (−1.8416) | (−1.8874) | (−1.0314) | (−1.2268) | (−2.4290) | (−2.0053) |
| Size | −0.0342 * | −0.0079 | −0.0841 *** | −0.1655 ** | −0.0338 | −0.0480 |
| | (−1.8918) | (−0.2301) | (−2.7881) | (−2.3902) | (−1.5830) | (−1.1521) |
| Lev | 0.9894 *** | 2.0819 *** | 0.9758 *** | 2.3333 *** | 0.3636 *** | 0.7434 *** |
| | (11.1932) | (13.3219) | (7.3706) | (7.7997) | (3.7596) | (3.9743) |
| Growth | 0.0202 | 0.0504 | −0.0442 | −0.1534 | 0.0751 ** | 0.1390 ** |
| | (0.6806) | (0.9913) | (−0.9141) | (−1.3222) | (2.3966) | (2.3765) |
| TobinQ | 0.0528 *** | 0.1204 *** | −0.0276 | −0.0704 | 0.0295 ** | 0.0585 ** |
| | (3.8022) | (5.0218) | (−1.2006) | (−1.3279) | (2.0073) | (2.1336) |
| Pattern | −0.1703 *** | −0.3013 *** | −0.1617 *** | −0.4000 *** | −0.2931 *** | −0.5939 *** |
| | (−5.0376) | (−4.8986) | (−3.1264) | (−3.3465) | (−7.7505) | (−7.7697) |
| Top10 | −0.0047 *** | −0.0074 *** | −0.0034 ** | −0.0104 *** | −0.0047 *** | −0.0092 *** |
| | (−4.7117) | (−4.1203) | (−2.2673) | (−3.0363) | (−4.3606) | (−4.4356) |

**Table 15.** *Cont.*

| Variable | (1) | (2) | (3) | (4) | (5) | (6) |
|---|---|---|---|---|---|---|
| | Information Disclosure Fraud | | Corporate Fraud | | Manager Fraud | |
| | Freq | Fraud | Freq | Fraud | Freq | Fraud |
| BigFour | 0.1620 ** | 0.4840 *** | 0.3252 ** | 0.9659 *** | 0.1949 ** | 0.3947 ** |
| | (2.3837) | (3.5844) | (2.5222) | (2.7452) | (2.5596) | (2.5493) |
| Anarpt | −0.0053 *** | −0.0132 *** | −0.0007 | −0.0021 | −0.0011 | −0.0033 ** |
| | (−6.7763) | (−8.3979) | (−0.5664) | (−0.7494) | (−1.3694) | (−2.1201) |
| Turnover | −0.0001 * | −0.0001 ** | 0.0000 | 0.0001 | −0.0000 | −0.0001 |
| | (−1.6935) | (−2.4004) | (0.2441) | (0.8498) | (−1.1464) | (−1.0566) |
| Cons | −1.8682 *** | −5.2950 *** | −1.8608 ** | −20.1411 | −1.4368 ** | −3.1671 ** |
| | (−4.0748) | (−4.7608) | (−2.4083) | (−0.0057) | (−1.9887) | (−2.2246) |
| Year | Control | Control | Control | Control | Control | Control |
| Industry | Control | Control | Control | Control | Control | Control |
| N | 22244 | 22244 | 22244 | 22244 | 22244 | 22244 |
| PseudoR2 | 0.0646 | 0.0820 | 0.0582 | 0.0625 | 0.0400 | 0.0421 |

Notes: The *t*-values in parentheses, *** means the significance level is 1%, ** means the significance level is 5%, and * means the significance level is 10%.

## 8. Discussion

### 8.1. Main Findings and Comparison with Other Studies

The focus of most HSR papers examines its impacts on economic development by improved flows of information, personnel, and capital [1,2,5]. Existing studies pay more attention to the impact of the opening of HSR on the macroeconomy and regional economy. A few studies examine HSR effects on firms and how improved transport alleviates information asymmetry and affect micro-enterprise behavior [6,45]. These papers provide support for our research. Based on previous studies, we found that the opening of high-speed rail has an inhibitory effect on corporate fraud. Specifically, this paper studies the impact of HSR on corporate behavior. The results show that: first, the HSR can inhibit the tendency and frequency of corporate fraud. The results are also significant after a series of robustness tests, such as the PSM-DID test, the placebo test, adding other transportation infrastructure as control variables, controlling the fixed effect at the company level, expanding the sample range, and eliminating the sample of big cities. Second, our results examine the transmission mechanism. We show HSR openings improve a firm's external supervision level, ease the financing constraints faced by the company, and then restrict corporate fraud. These results also support the research of relevant literature in other HSR fields, that is, the opening of high-speed rail can improve regional accessibility, alleviate information asymmetry [45], reduce the cost of cross-regional personnel mobility [46], and improve the efficiency of resource allocation [47]. Third, we document that market competition and company internal control affect the relationship between HSR openings and corporate fraud. The impact of HSR opening on corporate fraud is more significant at a lower level of market competition and internal control. Fourth, after distinguishing the types of fraud, we demonstrate that HSR openings have a significant impact on information disclosure fraud and manager fraud but no noticeable impact on operation fraud.

### 8.2. HSR Opening and the Sustainability of Capital Market

Substantial research demonstrates the indispensable link between capital markets and economic development [48,49]. The capital market plays a decisive role in resource allocation and provides an important channel for direct financing of enterprises. Therefore, the healthy and sustainable development of the capital market is the guarantee for the smooth operation of the economy. In recent years, China's capital market has developed rapidly. The number of A-share listed companies has grown from 13 in 1990 to more than 4100 in 2020.

However, with the continuous development of the capital market, corporate fraud has skyrocketed. The industry and academia are increasingly aware that corporate fraud seriously endangers the healthy and sustainable development of the capital market. When corporate fraud occurs, investors in the capital market are transmitting misleading information, which not only damages the interests of investors but also distorts the resource allocation efficiency of the capital market and destroys the market order [38]. In addition, we show that HSR introduction has a significant inhibitory effect on corporate fraud. Through the construction of a multiple regression model, a series of robustness tests, intermediary effect tests, and mechanism tests, the results show that the opening of HSR has restrained corporate fraud through strengthening external supervision and alleviating financing constraints. Based on the above analysis, we can draw the conclusion that the opening of high-speed rail can ensure the healthy development and sustainability of the capital market by restricting corporate fraud.

### 8.3. Policy Suggestion

This study makes a valuable supplement to the relevant literature on HSR opening and corporate fraud. At the same time, the following policy enlightenment can be obtained from the research conclusions of this paper. One, for the relevant government departments, we suggest continuing to increase the construction of transportation infrastructure. Our research results show that opening the HSR can restrict corporate fraud and standardize the capital market operation. In the future, relevant government departments can strengthen the construction of HSR and other transportation infrastructure, which has played a positive role in promoting the sustainability of China's capital market. Two, for regulators, we suggest that regulators implement differentiated supervision for companies with different levels of external supervision and financing constraints. Regulators should focus on whether companies with weak external supervision levels, strong financing constraints, low external market competition, and poor internal control levels have fraud. This targeted supervision will help to improve supervision efficiency and reduce supervision costs. Three, we suggest that regulators could further improve the information disclosure system of listed companies. Our statistical results show that information disclosure fraud accounts for the highest proportion among various types of fraud. Therefore, the regulators should focus on the supervision of corporate information disclosure, improve the information disclosure system of listed companies, and the information transparency of the capital market to ensure the standardized operation of the capital market.

### 8.4. Is China a Particular Framework?

Since the reform and opening-up, China's transportation infrastructure has developed rapidly and become a major boost to economic growth, especially the construction of HSR. In 2008, China opened the first formal HSR-Beijing Tianjin intercity HSR. Over the next ten years, China developed the largest HSR network in the world. By the end of 2019, China's high-speed rail has reached 35,000 km in operation, transporting 2.358 billion passengers throughout the year, and accounting for 64.4% of the total passenger volume in China. Therefore, we believe that it is necessary and important to study the opening of HSR in the context of China for the development of the capital market. Many existing studies on the upgrading of infrastructure such as HSR are also carried out in the context of China. For example, Gao et al., based on the opening of China's HSR, found that the upgrading of transportation infrastructure helps stimulate enterprise innovation [7].

Based on the data sample of China, we found that the opening area of HSR can promote economic development and the healthy operation of the capital market, but we cannot guarantee that our results can be applied to other countries in the world. This is because we conducted research based on Chinese samples and data. Whether there are similar conclusions in other countries requires further data collect and consideration regarding the political, economic, and cultural factors of other countries to carry out more empirical tests. This is beyond the scope of this paper, and it is not the main problem we want to study in

this article. Nevertheless, we believe that the conclusions of this paper still have reference value for government departments of other countries. Our results show that the opening of HSR accelerates the flow of personnel, information, and capital. This improves the capital market and affects the company's behavior and decision-making, which provides more empirical evidence for the economic consequences of transportation infrastructure upgrading. Our findings are consistent with Loughran and Tim, who indicate that in more remote areas, companies have less attention from investors, analysts, and the media [12]. Therefore, we think it is reasonable to speculate that the upgrading of transportation infrastructure in different countries in the world will promote the healthy development of the capital market to a certain extent. Relevant departments of other countries can learn from our experience and evidence, improve the transportation infrastructure, accelerate the resource circulation of the capital market, and then ensure the healthy, sustainable development of the capital market.

**Author Contributions:** Conceptualization, C.W.; methodology, C.W.; software, C.W.; validation, C.W., L.Z. and J.S.; formal analysis, C.W. and J.S.; investigation, C.W.; resources, C.W.; data curation, C.W.; writing—original draft preparation, C.W.; writing—review and editing, C.W., J.S. and L.Z.; visualization, C.W.; supervision, C.W.; project administration, L.Z.; funding acquisition, J.S., C.W. All authors have read and agreed to the published version of the manuscript.

**Funding:** This research was funded by the "Postgraduate Innovation Project of Beijing Jiaotong University" (2021YJS054), China and the National Social Science Foundation research project "the Reali-zation Path of Supporting the Comprehensive Construction of a Modern Country with Strong Transportation Network" (21AZD019).

**Conflicts of Interest:** The authors declare no conflict of interest.

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
