# Peer review of "High-Speed Railway Opening and Corporate Fraud"

_sustainability, doi:10.3390/su132313465_

Round 1
Reviewer 1 Report
Thank you for inviting me to review this manuscript, titled “High-Speed Railway Opening and Corporate Fraud”. The aim of this study is to explore the impact of high-speed railway opening on corporate fraud and to analyze its mechanism. The research topic proposed by the authors is original, relevant and interesting. The specific issue is defined.
Please find my detailed comments below:
First, I would suggest a very careful copy edit of the paper – there are a number of usage, grammar, and sentence structure errors. Also check: lines 236-237: 82.7% instead of 84.27%; line 265 ‘the propensity score matching method to match’, line 444 ‘the low market competition group’ instead of ‘low internal control level’, line 445 ‘low external supervision’ instead of ‘internal’.
The hypothesis is too general. It could be developed in order to make it more appropriate in relation to the objectives and context of the research. Or, issuing and testing a hypothesis is not mandatory. If research addresses a relatively new subject, then an exploratory attitude is more appropriate than testing a hypothesis.
The research methodology used by the authors is well grounded. Some details on control variables may be relevant: significance, impact.
Furthermore, several aspects can be included in the debate. Is there a link between corporate fraud, healthy development of the capital market, that depends (in this context) on high-speed railway opening, and sustainability? Is China a particular framework? Can results be applied to other countries? Can relevant government departments in other countries stimulate the construction of high-speed railways and other transport infrastructure to ensure a healthy development of the capital market?
No reference is made to whether or not the results are in line with previous research.
Thank you for this interesting paper.
Author Response
"Please see the attachment."

Reviewer 2 Report
First of all, I would like to congratulate the authors for the research carried out, which shows the detail and interest with which it has been done. I note that the structure followed is adequate and easy to understand. The novelty and interest of the work is excellent which shows that it has been a well worked research.
The objective of this paper is to explore the impact of high-speed railway opening on corporate fraud and analyze its mechanism. However, after reading and analyzing the research, it only remains for me to thank you for allowing me to analyze this research and to advise some points to improve.
- The plagiarism rate is 19%. Perhaps it would be interesting to reduce it a little more since the journal in which it is intended to be published is of high impact.
- For the same reason, as it is a high impact journal, I advise to include in the literature review some reference to Sustainability, since there are many publications in the journal focused on this topic.

Reviewer 3 Report
The current paper titled, 'High-Speed Railway Opening and Corporate Fraud' provides a valuable addition to the the extant literature on corporate fraud. It is an interesting paper as it adds a new perspective to empirical research on corporate misconduct broadening the empirical literature from firm-specific factors only to add a macro perspective. I have few suggestions for the authors.
1) First, I think the paper should go through extensive language correction and editing.
2) The state of art of fraud research needs further elaboration. I suggest authors to add more anecdotal evidences to empirical and theoretical literature on corporate misconduct and illicit activities of the firms.
3) A graph or a short table guiding the readers about the order of presentation of findings might be helpful.
4) The results presentation is great, but there is dearth of supporting empirical studies, especially in the discussion of results.
5)It would be really helpful for the readers to follow the fluidity if the authors could elaborate the purpose of doing a number of robustness testing.
Reviewer 4 Report
Dear Authors,
First of all, congratulations on the work presented in your article.
The structure of the article, and methodology adopted, seems to be adequate to the research developed.
The research topic is interesting.
After reading your article wasn’t clear the connection of the journal scope. In my opinion, this is the weakest point of the article. The link to the scope of the journal should be improved. The authors should added some references concerning to sustainability and, connected to capital market.
In the conclusions i didn't see any comment concerned to sustainability, healthy and stable development to capital market.
The article has some errors. For example, please see line 211 "ananpt" should be "anarpt".
Best of luck with your article!
